# Biochar as a Sustainable Component of Low-Emission Building Materials

**DOI:** 10.3390/ma18173941

**Published:** 2025-08-22

**Authors:** Iwona Ryłko, Robert Zarzycki, Łukasz Bobak, Paweł Telega, Agnieszka Didyk-Mucha, Andrzej Białowiec

**Affiliations:** 1Department of Applied Bioeconomy, Wrocław University of Environmental and Life Sciences, 37a Chełmońskiego Str., 51-630 Wrocław, Poland; iwona.rylko-polak@upwr.edu.pl (I.R.); lukasz.bobak@upwr.edu.pl (Ł.B.); pawel.telega@upwr.edu.pl (P.T.); 2Department of Advanced Energy Technologies, Czestochowa University of Technology, 73 J.H. Dąbrowskiego St., 42-200 Częstochowa, Poland; robert.zarzycki@pcz.pl; 3Selena Industrial Technologies, Pieszycka 1, 58-200 Dzierzoniów, Poland; agnieszka.didyk@selena.com

**Keywords:** biochar, pyrolysis, wood chips, carbon footprint, low emission, VOC, LCA

## Abstract

Biochar (BC), derived from wood biomass through pyrolysis, exhibits properties that make it a promising additive in mortars for sustainable construction. This study investigated the influence of biochar produced at three pyrolysis temperatures (450 °C, 550 °C, and 700 °C) on the performance of cementitious adhesive mortars. The evaluation encompassed physicochemical characterization, mechanical and adhesive strength, volatile organic compound (VOC) emissions, leachability of contaminants, and a life-cycle assessment (LCA). The results demonstrate that biochar obtained at 700 °C has the highest carbon content, an alkaline pH, and increased porosity. In contrast, biochar produced at 450 °C exhibits better sorption capacity and a higher concentration of functional groups. Incorporating 1–5% BC (produced at any temperature) improves mortar performance; however, higher doses negatively affect adhesion to expanded polystyrene board (EPS) and concrete. Emissions of VOCs and leachable metals largely remained within environmental threshold values, with only isolated instances of exceedance. The LCA revealed that substituting mineral fillers with biochar could reduce the carbon footprint by up to 35% compared to the reference formulation. These findings confirm biochar’s potential as a safe and environmentally beneficial component in low-emission construction materials, aligning with the principles of the circular economy and climate-neutral goals.

## 1. Introduction

Biochar, obtained by pyrolysis of wood chips, is a promising material in the context of sustainable development and environmental protection [1], thanks to its unique properties, such as its high carbon content, porous structure, and ability to store carbon dioxide. Biochar is gaining more and more recognition in various sectors, including construction. The process of its production consists of the thermal decomposition of biomass under conditions of limited oxygen access, which allows the transformation of organic material into a stable carbon product [2]. A particular advantage of biochar is that it can be produced from renewable raw materials, such as wood waste, which often remain unused, generating only disposal costs. In the context of the circular economy, biochar not only reduces waste but also promotes carbon sequestration, making it a so-called “carbon negative” material [3].

Wood chips, which are the waste of the wood industry and forest management, are a good raw material for biochar production [1]. They are characterized by wide availability, low price, and appropriate chemical composition, conducive to an effective pyrolysis process. Poland, as one of the leaders in the European wood industry, generates approximately 10,764 mln m^3^ of wood chips that could be converted into biochar for the construction industry [4]. In this way, it is possible not only to reduce waste but also to develop innovative and ecological solutions to support low-carbon construction. The inclusion of biochar in the production of building materials such as cement mortars has the potential to significantly reduce the carbon footprint of the construction sector, which currently accounts for a significant share of global CO_2_ emissions [5].

In addition, the specific properties of biochar, such as porosity, high specific surface area, and chemical stability, make it a good additive for cementitious composites. It improves the mechanical and thermal properties of materials while contributing to their durability and resistance to environmental factors [6]. The biochar production process is part of a sustainable development strategy, minimizing the negative impact on the environment and promoting innovative technologies that support energy efficiency and the reduction of greenhouse gas emissions. In this way, biochar combines ecological and technological goals, providing a comprehensive solution for the construction sector [7]. The purpose of this research is to comprehensively evaluate the effect of the addition of biochar, extracted from the pyrolysis of wood chips carried out at different temperatures (450 °C, 550 °C, and 700 °C), on the physicochemical, mechanical, and environmental properties of thermal mortars used in construction.

In recent years, biochar has garnered attention as a sustainable additive in cementitious composites, owing to its potential to enhance environmental performance and reduce carbon emissions. For instance, small additions of biochar (1–2 wt%) have been shown to adjust the water-to-cement ratio beneficially, increasing compressive and flexural strength by up to 20–50% in mortar and concrete applications [8]. A recent meta-analysis of over 600 case studies reported that plant-based biochars—especially those produced above 450 °C—improved the 28-day compressive strength of Portland cement composites by 3–13% [9].

However, despite these benefits, several studies have identified performance limitations. Patel and Stobbs [10] reported that certain litter-derived biochars led to decreased strength due to dilution effects and poor hydration product formation. Other research highlighted that high replacement levels (>5%) or pyrolysis at suboptimal temperatures (e.g., low-temperature, high-volatile biochars) could lead to strength reduction, delayed setting times, and increased water demand [11,12]. Moreover, variability in feedstock type, pyrolysis conditions, particle size, and surface chemistry has been shown to contribute to inconsistent results across studies [13].

These mixed findings underscore the need for a more systematic evaluation framework. While biochar’s environmental advantages are well-recognized, relatively few studies have addressed the trade-offs between mechanical performance and environmental safety, particularly in the context of heavy metal leachability or volatile compound emissions. Furthermore, integrated studies combining multi-temperature analysis with comprehensive environmental assessment tools—such as life-cycle assessment (LCA)—remain scarce [14].

This work addresses these research gaps by investigating biochar produced at three distinct pyrolysis temperatures (450 °C, 550 °C, and 700 °C) and evaluating the impact on the physicochemical, mechanical, and environmental properties of thermal mortars. This includes elemental composition, surface area, porosity, mechanical performance (e.g., compressive and flexural strength), as well as environmental safety indicators (VOC emissions and pollutant leachability) and full cradle-to-gate LCA. The research aims to identify the most balanced solution—particularly whether biochar produced at 550 °C offers an optimal compromise between material performance and environmental compliance. The results contribute to expanding knowledge on the responsible and scalable use of biochar in the construction sector, supporting both circular economy principles and climate goals.

## 2. Materials and Methods

### 2.1. Materials

#### 2.1.1. Biochar

Biochar was produced from wood chips. The wood chips used in the study came from northeastern Poland, mainly from the Podlasie region (around Białowieża and Hajnówka). The raw material was a mixture of coniferous (Scots pine and spruce) and deciduous (birch, alder, and oak) wood, typical of the forest management of the area. The material included both remnants of post-production sawmills and nursing cuts carried out by the RDSF Białystok forest districts. The wood chips were subjected to a pyrolysis process at three different temperatures: 450 °C, 550 °C, and 700 °C. The study used the biochar produced at these three pyrolysis temperatures, which allowed for the assessment of the impact of biomass thermal decomposition conditions on the properties of biochar and its effect on cement mortars. This choice was based on the literature, according to which the pyrolysis temperature significantly affects the structure, chemical composition, and functional properties of biochar [5].

Biochar obtained at a lower temperature (450 °C) is characterized by a higher content of oxygen functional groups (–OH and –COOH), which increases its chemical reactivity and sorption capacity, but at the same time leads to higher emissions of volatile organic compounds (VOCs). A temperature of 550 °C is a compromise between preserving functional groups and achieving greater chemical and structural stability [5]. On the other hand, biochar burned at 700 °C is characterized by a high degree of carbonization, alkaline pH, and a well-developed porous structure, which improves the thermal insulation properties and durability of the material while reducing VOC emissions [5].

This temperature range has been repeatedly confirmed in the literature as representative of the differentiation of biochar properties depending on the conditions of its production.

Before the pyrolysis, wood chips were ground to a fraction of approximately 0.2–6 cm and dried in a POL ECO SLW 115 laboratory dryer to a moisture content of approximately 20%.

The biochar used in the study was obtained by pyrolysis of wood chips in a specialized acid-resistant steel tube reactor with a volume of 10 dm^3^. The reactor was placed in an electric furnace with automatic temperature control and an inert gas flow system [5,7].

In each cycle, 500 g of dry wood chips were fed into the reactor. The pyrolysis process was carried out at the three target temperatures—450 °C, 550 °C, and 700 °C—to vary the physicochemical properties of the biochar. The temperature rise rate was 10 °C/min. After reaching the target temperature, the material was kept in the furnace for 60 min (isothermal retention time), which allowed for full stabilization of the carbon structure [2,5,7].

Throughout the process, an inert gas—technical nitrogen (99.99% purity)—was introduced into the reactor at a flow rate of 120 cm^3^/min. The aim was to eliminate oxygen access and prevent uncontrolled combustion of the biomass. The nitrogen flow was maintained during both the heating and cooling phases of the reactor [7].

After the pyrolysis process was completed, the biochar samples were left to cool in a nitrogen atmosphere, then weighed and sieved to remove dusty fractions.

#### 2.1.2. Mortar Components

For this study, proprietary adhesive mortar formulations were developed [15] with Portland cement grade CEM II 42.5 A-V as the primary ingredient, accounting for approximately 30% of the composition of each formulation. This cement belongs to the group of materials with reduced carbon-dioxide emissions during production, making it compliant with current European Union regulations promoting the use of low-carbon building materials [16]. In Poland, CEM II grade cement is increasingly being chosen in response to stricter environmental standards and growing pressure to reduce the carbon footprint of construction [17].

The developed mixes also included quartz sand of 0.1–0.5 mm fraction as a filler, the proportion of which varied depending on the formulation variant [8]. A cellulose ether—hydroxy–propyl–methylcellulose with a viscosity of approximately 45,000 mPa-s—was used in an amount of approximately 0.3% to ensure adequate stabilization of the mixture [15]. In addition, 1% of a re-dispersible polymer based on ethylene–vinyl acetate copolymer was introduced, which acted as an agent to improve the adhesion, flexibility, and resistance to cracking of the mortar [15].

Biochar obtained by pyrolysis of wood chips conducted at three different temperatures—450 °C, 550 °C, and 700 °C—was introduced into the mixture prepared in this way. These additives were used in three quantitative variants—1%, 5%, and 10% by weight—in order to evaluate their effects on the physicochemical, mechanical, and environmental properties of the adhesive mortars (Table 1).

### 2.2. Methods

#### 2.2.1. Experiment Design

In the first stage of the study, a detailed physicochemical characterization of the biochar produced by pyrolysis of wood chips at 450 °C, 550 °C, and 700 °C was carried out. Among others, elemental composition (CHNS), heat of combustion, and pH, as well as leachability of pollutants and emissions of volatile organic compounds (VOCs), were evaluated. These analyses were aimed at identifying the optimal properties of biochar in terms of their suitability as an additive for cement mortars.

In the next stage, adhesive mortars previously containing biochar were prepared based on the formulations developed.

#### 2.2.2. Experiment Procedure

The developed mixtures were subjected to a mixing process. This process was carried out in three steps:
1.Homogenization of dry components. All dry components, including cement, sand, cellulose ether, polymer, and an appropriate amount of biochar, were poured into the mixer and mixed for 30 s to pre-homogenize the mixture.2.Adding water and mechanical mixing. Water was gradually added to the dry mixture while stirring with a mechanical mixer at 600 rpm for 90 s. The goal of this phase was to achieve even wetting of the ingredients.3.Rest phase and re-mixing. After completion of the first mixing, the mixture was allowed to rest for 2 min, after which it was mixed again for another 60 s to ensure complete homogeneity and absence of lumps in the finished mortar [18].

Evaluation of the consistency of the prepared mixtures was carried out using the spreading table method in accordance with PN-B 04500 [18], allowing comparison of rheological properties among different formulation variants.

#### 2.2.3. Analytical Methods

Analysis of C, H, N, S, and O of biochar pyrolyzes at 450 °C, 550 °C, and 700 °C.

The analyses of C, N, H, S, and O content in biochar were carried out by elemental analysis [6] by the combustion method (Dumas), which made it possible to accurately determine the content of elements in biochar following ASTM D5373 [19]. The samples were obtained using the Perkin Elmer PE2400 series II CHNS/O analyzer (PerkinElmer, Inc., Shelton, CT, USA).

Adhesion and mechanical strength tests of cementitious mixtures containing tested biochar.

Adhesion to EPS (exposed polystyrene) and concrete according to EAD 040083-00-0404 [20] is aimed at assessing the effectiveness and durability of the adhesive bonding with the substrate. They are performed as part of the technical assessment for ETICS systems.

Description of the adhesion test according to EAD 040083-00-0404 [20]:
Adhesion to concrete: Samples are made with adhesive mortar applied to the concrete substrate. After 28 days of curing, a pull-off test is carried out following EN 1542:1999 [21]. The result is given in MPa and documents the type of destruction (adhesion, cohesion, mixed). Minimum adhesion value: usually ≥ 0.25 MPa (or according to the design requirement).EPS adhesion: The adhesive is applied to polystyrene boards in a standard arrangement (e.g., using the band–point method). After drying, a tear test is carried out using steel plates glued to the sample. Continuous contact and no detachment within the adhesive layer are required—usually, a break in the EPS itself is desirable (which indicates good adhesive adhesion).Mechanical strength: Tests of mechanical strength of mortars according to EN 998-1:2016-12 apply to mortars intended for masonry and plastering applications (plaster and masonry mortars) [22]. Mechanical evaluation is a key element in the classification of these mortars and primarily includes compressive strength and flexural strength.

1.Compressive strength [22]:

It is performed on samples (usually cuboidal or cubic).

The samples are formed and matured for 28 days under laboratory conditions (20 °C, 65% humidity). The result is given in MPa (megapascals). Depending on the mortar class, it is assigned to categories (np. CS I, CS II, CS III, or CS IV):
-CS I: 0.4–2.5 MPa-CS II: 1.5–5.0 MPa-CS III: 3.5–7.5 MPa-CS IV: > 6.0 MPa

2.Flexural strength [15]:

As a rule, it accounts for approximately 1/3 to 1/5 of the compressive strength. The result is expressed in MPa.

Consistency determination using the Navikov cone by precipitation method [18,23].

This test is used to determine the consistency of the mortar (i.e., its “working density”) and consists of measuring the precipitation of the Navikov cone in fresh mortar. It is a simple, fast, and commonly used method, especially in building materials laboratories [18,23].

Testing the leachability of contaminants.

In order to assess the potential environmental impact of biochar and other waste materials used as secondary raw materials in cement mortars, leaching studies were carried out. This test determines whether the materials meet the criteria for release for use or storage following applicable legislation.

The research procedure was carried out following the Regulation of the Minister of Economy of 16 July 2015, Annex 2 [24], concerning the conditions for accepting waste at landfills. The European standard PN-EN 12457-4:2006 [25] was used as the test methodology, which describes the procedure for the leachability test in a solid–liquid system at the ratio of L/S = 10 dm^3^/kg (liters of extraction liquid per kilogram of dry material).

The range of parameters determined included, e.g., heavy metals (such as cadmium, lead, mercury, nickel, zinc, copper), inorganic anions (chlorides, sulfates, fluorides), general parameters (pH, electrolytic conductivity, total organic carbon—TOC), and other potentially harmful compounds identified in the analyzed samples.

The results were compared with the permissible limit values specified in the regulation, which made it possible to assess the suitability of the tested materials for further use in construction, from the point of view of their environmental safety.

Measurement of volatile organic compounds (VOCs) emissions.

The emission of volatile organic compounds (VOCs) from cement mortars was assessed using gas chromatography coupled with mass spectrometry (GC-MS) [26,27]. Analyses were performed with an Agilent GC-MS system (GC 7890B/MS 5977B) equipped with a DB-5MS column (30 m × 0.25 mm × 0.25 μm; Agilent, Santa Clara, CA, USA). VOCs were extracted using a FIB-C-WR-95/10-P1 Carbon WR/PDMS fiber (PALSystem, CTC Analytics AG, Zwingen, Switzerland) installed in a Gerstel MPS robotic autosampler (Gerstel GmbH & Co. KG, Mülheim an der Ruhr, Germany). A 2.5 ppm solution of caryophyllene was used as the internal standard. The fiber was preconditioned at 270 °C for 30 min and additionally heated for 1 min before and after each run. Samples were conditioned at 60 °C for 10 min, followed by VOC extraction at the same temperature for another 10 min, and analytes were desorbed in the injector for 5 min. Instrument settings included an injector temperature of 250 °C, a split ratio of 10:1, and a helium flow rate of 1.0 mL·min^−1^. The oven temperature program started at 40 °C (2 min hold), ramped to 150 °C at 5 °C·min^−1^, and then to 250 °C at 10 °C·min^−1^. The quadrupole, ion source, and transfer line temperatures were set to 150 °C, 230 °C, and 250 °C, respectively [28]. Compound identification was performed using the NIST17 spectral library [29].

Carbon footprint and life-cycle analysis.

The carbon footprint was estimated using life-cycle assessment (LCA) software (One Click LCA Ltd., Helsinki, Finland, One Click LCA software, https://www.oneclicklca.com (accessed on 20 February 2025)). The objective of the analysis was to determine the annual CO_2_ emissions associated with a reference formulation and its variants incorporating waste-derived raw materials such as biochar, fly ash, and sodium lignosulphonate. The calculations were based on a cradle-to-gate approach, encompassing emissions generated during the production and delivery of raw materials to the manufacturing facility.

All formulation variants contained a constant cement content (30%). Variations occurred in the proportions of sand, ash, lignosulphonate, and polymer additives (cellulose ether and polymer dispersions). Emission factors were sourced from the Ecoinvent database and Environmental Product Declarations (EPDs). For each formulation, the carbon footprint was calculated by multiplying the proportion of each ingredient by its corresponding emission factor. The total emissions for 100 kg of mixture were summed, and annual emissions were then estimated using the following equation:$$Carbon\;footprint\;[MG\frac{{CO}_{2}}{year}]=\frac{emission\;per\;100kg\;\times \;total\;annual\;mass\;\left[kg\right]}{100\;\times \;1000}$$

Based on the emissions data, a life-cycle assessment was also conducted following ISO 14040/14044 [30] and ISO 14067:2018 [31], focusing on the environmental impacts of the developed mixtures. The analysis followed the “cradle-to-gate” scope and included the following modules:
A1—Raw material extraction and supply.A2—Transport to the plant and energy used during mixing.A3—Emissions associated with manufacturing processes.

The LCA included not only the calculation of the carbon footprint but also the evaluation of raw material and energy consumption, their effects on energy efficiency, and broader environmental impacts such as greenhouse gas emissions, water use, and potential material toxicity [32,33].

The research methods used allowed for a comprehensive assessment of the physicochemical properties of the biochar itself and their potential applications in cement mixtures.

## 3. Results and Discussion

### 3.1. Mechanical and Technological Performance of Cement Mortars

Table 2 below shows the test results for biochar samples pyrolyzed at the three different temperatures.

For an in-depth interpretation of the results, the experimental data obtained on the mechanical and environmental properties of adhesive mortar mixtures with biochar additives were related both to the applicable building standards and to the research results reported in the scientific literature.

All mortar variants met the minimum requirements of EN 998-1:2016 [22] and the EAD guideline 040083-00-0404 in terms of technical performance [20]. This means that the adhesion to concrete substrates exceeded the threshold value of 0.25 MPa, and the adhesion to EPS was at least 0.08 MPa. The highest results were achieved in samples containing biochar fired at 550 °C and 700 °C, indicating a positive effect of the higher pyrolysis temperature on mortar adhesion.

The trends observed in this study are in agreement with the results reported in current scientific publications. A study by Ali D. et al. [32] showed that an increase in pyrolysis temperature leads to an increase in the carbon content and a decrease in the oxygen content of the biochar structure, resulting in greater chemical durability and improved mechanical properties of cementitious composites.

Similar conclusions were drawn from the work of Maljaee et al. [33], where the optimum biochar content in cement mortars ranged from 1% to 5%, with beneficial effects particularly evident for biochar fired at 550 °C. These authors highlighted the role of porosity and the specific surface area of biochar as factors favoring better bonding to the cement matrix.

Murali et al. [34], on the other hand, warn against using too high doses of biochar (above 10%), which can have a diluting effect on the binder and lead to a weakening of the mechanical structure—an effect that was also observed in this study in formulation variants F3, F6, and F9.

In terms of environmental performance, high-temperature biochar (above 600 °C) exhibits reduced leachability and increased chemical stability due to stronger immobilization of heavy metals and decreased volatility of organic compounds. These properties are advantageous in cement-based materials, as they correlate with lower fugitive emissions and improved pH stability in mortars containing 700 °C biochar [35].

It is also worth citing the work of Tan et al. [36], which showed that biochar obtained at approximately 550 °C provides the best balance among mechanical stability, rheological workability of the mortar, and its environmental performance.

The observed differences among the different mortar variants can be explained based on the structural changes occurring in the biochar under the influence of the pyrolysis temperature. At lower temperatures (450 °C), biochar is formed that is richer in oxygen and volatile compounds, which increase reactivity but reduce chemical durability and increase leachability. At higher temperatures (700 °C), biochar becomes more stable, alkaline, and resistant to degradation, but at higher concentrations can lead to brittleness and over-saturation of the mineral matrix with ash [37,38]. As reported by Yang et al. (2025), biochar enriched with isolated Ca–Nx sites has the potential to boost Fe-catalyzed Fenton-like oxidation of organic pollutants through improved catalytic activity and optimized surface chemistry [39].

A temperature of 550 °C proved to be the most optimal. The biochar obtained had balanced properties: moderate porosity, low VOC emissions, appropriate pH, and a positive effect on the mechanical properties of the mortars.

### 3.2. Mechanical Assessment of Cement Mortars

All measurements were performed in triplicate, and the results are presented as mean values with standard deviations (±0.02 MPa).

### 3.3. Test Results According to EAD040083-00-0404 and EN 998-1:2016-12

Table 3 provides a summary of the formula tests from Table 2.

### 3.4. Water/Cement Ratio (w/c)

The reference sample (F0) had the lowest w/c ratio, typical of high-strength mortars. The following changes were observed for mixtures with biochar: For biochar fired at 450 °C (F1–F3), the ratio remained at 0.22, indicating stability and no significant effect on water requirements. The 550 °C biochar (F4–F6) caused values to fluctuate between 0.21 and 0.26, with the F4 sample showing a markedly increased demand (0.26), which may suggest a higher absorption or chemical influence. For biochar fired at 700 °C (F7–F9), a gradual increase in w/c from 0.22 to 0.24 was observed, confirming the trend of increasing water retention with growing proportions of high-temperature biochar. Despite these changes, the consistency of the mixtures remained stable.

### 3.5. Adhesion to EPS

The results of adhesion tests to polystyrene foam showed that samples F1, F2, F4, F5, and F6 met the required minimum (≥0.08 MPa). The reference sample F0 reached the limit (0.10 MPa), while samples F3, F7, F8, and F9 obtained results below the required threshold—the lowest result was obtained for F9 (0.05 MPa). The analysis indicates that excessively high doses of biochar (above 5%) negatively affect adhesion to EPS, which may be due to a reduction in the bonding surface or an altered microtexture of the mortar. The best results were obtained for samples F1 and F4, each containing 1% biochar.

While this study evaluated adhesion strength using standardized pull-off testing, further microstructural analyses (e.g., SEM of the interfacial transition zone) are recommended to better understand the interaction mechanisms between the mortar matrix and different substrates such as EPS and concrete.

### 3.6. Adhesion to Concrete

In terms of adhesion to the concrete substrate, all samples F0 to F6 met the requirements of the standard (≥0.25 MPa). Samples with 700 °C BC added at higher concentrations (F7–F9) did not meet the criteria, suggesting reduced adhesion with excess high-temperature biochar. Sample F1 achieved the best result (0.50 MPa), confirming the beneficial effect of a small BC addition on bonding to mineral surfaces.

### 3.7. Flexural Strength

All tested samples showed satisfactory flexural strength values, ranging from 5.2 to 6.4 MPa. The highest results were achieved for samples F0, F5, and F6 (~6.2–6.4 MPa), while the lowest results were achieved for F3 and F4 (approximately 4.6 to 5.2 MPa). The presence of biochar additives in the amount of 5–10% can slightly weaken the bending resistance, with the effect depending on the pyrolysis temperature and the amount of BC in the mixture.

### 3.8. Compressive Strength

All analyzed formulations met the requirements of EN 998-1:2016-12 [22] for minimum compressive strength (≥20 MPa). The reference samples F0 and F7 (biochar 700 °C, 10%) reached the highest values (25.5 MPa). The lowest strength was recorded for sample F4 (20.5 MPa), containing 1% BC from pyrolysis at 550 °C. The results indicate that the moderate addition of biochar does not significantly reduce the strength of mortars, although it may affect the microporosity and compactness of the structure depending on the BC production conditions.

The observed reduction in compressive strength for sample F4 (biochar at 550 °C, 1%) may be attributed to an unfavorable balance between particle reactivity and internal porosity. Biochars produced at intermediate temperatures may still contain residual volatiles or underdeveloped carbon structure, potentially leading to poor bonding with the cement matrix or microstructural disruption [40]. This suggests that not only temperature but also pyrolysis completeness and additive dispersion must be carefully controlled to ensure consistent mechanical performance.

### 3.9. Environmental Assessment of Cement Mortars

The environmental results for the developed formulations with different biochar content are collected below. Leaching tests were conducted in duplicate for each sample, and the reported concentrations represent average values; standard deviations are ±0.01–0.1 mg/kg at low concentrations.

#### Leaching of Contaminants

Ten waste samples, marked as F0–F9, were tested in order to assess the level of selected heavy metals and compare them with the permissible values in accordance with the applicable environmental standards. The analysis covered seven parameters: mercury (Hg), molybdenum (Mo), nickel (Ni), lead (Pb), antimony (Sb), selenium (Se), and zinc (Zn) (Figure 1).

All samples met the requirements for mercury, nickel, lead, and antimony concentrations. The measured content of these elements in each sample was significantly lower than the permissible limits (respectively: Hg—0.01 mg/kg, Ni—0.4 mg/kg, Pb—0.5 mg/kg, Sb—0.06 mg/kg).

In the case of molybdenum (limit 0.5 mg/kg), the F7 sample recorded a value of 3.1 mg/kg, which means a significant exceedance of the norm. Similarly, for zinc (limit of 4 mg/kg), the highest value of 4.7 mg/kg was obtained in the F0 sample, which also indicates that the permissible content was exceeded. Both cases may suggest a potential environmental risk and require further analysis.

Although biochar produced at 700 °C demonstrated superior performance in terms of mechanical strength and VOC reduction, its elevated molybdenum leachability indicates a potential environmental risk. Therefore, its application should be approached with caution, and further optimization of pyrolysis conditions or feedstock composition is recommended to ensure regulatory compliance and safe long-term use.

Selenium concentrations in the tested samples ranged from 0.002 to 0.075 mg/kg, with an acceptable limit of 0.1 mg/kg. The highest value was again recorded in the F0 sample, but it was still within the normal range.

Except for two exceedances—molybdenum in the F7 sample and zinc in the F0 sample—all other results follow the requirements specified for the leachability test (liquid/solid = 10 L/kg). Most samples can therefore be considered safe for heavy metal content, while the F0 and F7 samples should be further inspected or classified as potentially hazardous, following applicable waste-management regulations. The elevated molybdenum leaching in sample F7 (3.1 mg/kg) and zinc exceedance in sample F0 (4.7 mg/kg) suggest potential risks associated with certain biochar formulations. These values surpass the thresholds set for inert waste according to EU regulations, indicating the need for careful control of feedstock composition and pyrolysis conditions. While no acute ecological toxicity is expected at the scale of application, future research should include full environmental risk assessments using source–pathway–receptor models or leachate transport simulations to evaluate long-term impact and material compliance.

The content of selected chemical components related to the solubility and migration of pollutants in the environment was considered: chlorides (Cl^−^), fluorides (F^−^), sulfates (SO_4_^2−^), TDS (total dissolved solids), and DOC (dissolved organic carbon). Their levels were assessed in ten waste samples (marked F0–F9) and then compared with the permissible limits for the basic test (10 L/kg) (Figure 2).

The analysis indicated the following:
Fluorides: exceedances occurred in the F0 sample, where the concentration was 11 mg/L, exceeding the limit by 10%.Sulfates: samples F2, F3, F4, and F5 showed values well above the permissible limit (up to 3990 mg/L), indicating a high risk of sulfate leaching into the environment.DOC and TDS: all values were well below the accepted limits, suggesting a moderate presence of organics and total salts.Fluorides and sulfates have shown the greatest variability and potential exceedances, which require attention in the context of further landfilling.

The other parameters, including TDS and DOC, did not constitute an environmental hazard according to the available results.

To ensure environmental safety, further monitoring is recommended, especially for the F0 (fluoride) and F2–F5 (sulphate) samples.

### 3.10. Emissions of Volatile Organic Compounds

Biochar (BC) produced by the pyrolysis of lignocellulosic materials can significantly affect the chemical composition of the composite matrix, including the presence and nature of volatile compounds. This part of the study aimed to assess the effect of BC content and firing temperature on the VOC emission profile. Analyses were performed using mass detector gas chromatography (GC-MS), enabling the generation of total ion chromatograms (TICs) and the identification of the main compounds present in the samples (Figure 3). Typical uncertainty range: ±1–5 µg/m^3^ or ±2–10% of the measured value.

Description of samples:
F0—reference sample (without BC additive);F1–F3—samples with 1%, 5%, and 10% BC, fired at 450 °C;F4–F6—samples with 1%, 5%, and 10% BC, fired at 550 °C;F7–F9—samples with 1%, 5%, and 10% BC, fired at 700 °C.

Figure 3 below shows a comparison of TIC chromatograms for all samples analyzed. On the X axis, there is the retention time (min), and on the Y axis, the intensity of the TIC signal, corresponding to the total number of detected ions at a given moment of the analysis.

F0 (reference):
Straight profile, with a single dominant peak (~21 min RT).Limited presence of volatile compounds.

Formulation with biochar fired at 450 °C (F1–F3):
An increase in the number of peaks was observed, especially at RT ~28 and ~34 min.The lower pyrolysis temperature promotes the presence of more reactive and unstable organic compounds.As the BC (F9) content increases, the intensity of TIC increases.

Formulation with 550 °C fired biochar (F4–F6):
More intense peaks appeared in the RT ~28–34 min regions, indicating the formation of cyclic and condensation compounds.The BC content affects the intensity: F13 shows the most extensive profile in this group.

Formulation with 700 °C fired biochar (F7–F9):
The highest TIC intensities and the highest profile complexity.This indicates the presence of secondary products resulting from deep carbonization and aromatization.F6 (10% BC) was the sample with the highest volatile emissions.

Both the biochar content and the temperature of its firing significantly modify the profile of volatile compounds in the samples. With the increasing pyrolysis temperature and the participation of BC:
The total intensity of the TIC increases;New compounds appear in the higher RT ranges;The chemical composition becomes more complex.

The bar graph below (Figure 4) shows the relative area of the peaks (%) for three key retention times:
~21.3 min—dominant peak in each sample;~28.9 min—appears and strengthens with increasing BC content;~34.8 min—characteristic for samples with biochar fired at higher temperatures.

Three dominant chromatographic peaks have been identified, which show characteristic relationships concerning firing conditions and carbon content:
~21.3 min peak: Observed as dominant in all samples, regardless of their origin and thermal conditions. Its intensity confirms the presence of stable organic compounds formed at an early stage of pyrolysis, probably related to low-molecular volatile components typical of biomass.~28.9 min peak: Its surface area increases steadily from the F1 to F6 samples, reaching its maximum in samples with higher BC content. The compound represented by this peak may be an indicator of the presence of hard, partially aromatic compounds, characteristic of the average pyrolysis temperature range (approx. 400–600 °C).~34.8 min peak: Clearly appears only in F4 and later samples, and its intensity increases significantly in the F6–F9 samples. It is strongly correlated with the firing temperature, indicating the presence of highly condensed aromatic structures or polycyclic compounds. It can be regarded as a chemical marker of intense carbonization.

Quantitative analysis of the surface area beneath these peaks showed clear differences among the samples (Figure 4), allowing a qualitative assessment of their composition and processing level. Samples F5–F9, characterized by high intensities of all three peaks, represent an advanced level of carbonization and a high BC content. In contrast, the reference sample and F1–F2 exhibit trace peaks of ~34.8 min and low peak values of ~28.9 min, confirming their low carbon content and mild processing conditions.

In order to deepen the qualitative analysis of organic components present in biochar samples, the identified chemical compounds were classified according to their structural types [17]. Their relative intensity (%) was then estimated by comparing the reference sample and the sample sets F1–F3, F4–F6, and F7–F9, representing the increasing level of carbonization.

Figure 5 shows a comparison of the six main groups of chemical compounds:
Non-oxygenated compounds (CO_2_, C1–C4)—dominant in the reference sample, their share systematically decreases with the increase of the pyrolysis temperature, which indicates their high volatility and instability under carbonization conditions.Ketones and aldehydes—their proportion is highest in the F1–F3 samples and then decreases slightly in the F7–F9 samples. These compounds are typical products of hemicellulose and cellulose decomposition.Aromatic and polycyclic compounds (e.g., benzenes, furans)—their share increases significantly in the F4–F9 samples, indicating the intensification of aromatization processes at higher temperatures.Siloxanes (cyclosiloxanes)—these compounds, although often treated as technical impurities, also show a greater presence in the later stages of carbonization.Terpene compounds and esters—moderate content in all sample groups, slightly higher in the F7–F9 samples, which may indicate stable origin from the starting material (e.g., softwood).Fatty acids and condensation products—the share of this group increases significantly in F7–F9, which suggests the presence of compounds formed as a result of secondary condensation and the polymerization of hydrocarbons under high-temperature pyrolysis conditions.

The summary of chemical profiles (Figure 5) indicates the evolution of the composition of volatile products as a function of the pyrolysis temperature. As the carbonization intensity increases, there is a transition from light, saturated non-oxygenated compounds to more complex, condensed aromatic structures. Such a transformation confirms the high thermal reactivity of the biomass and the progressive nature of the formation of a biochar structure with greater durability and sorption properties [37].

Analysis of volatile compounds (VOCs) present in biochar samples showed significant differences in chemical composition depending on the degree of carbonization, which was unequivocally confirmed by GC-MS data. The samples subjected to a higher pyrolysis temperature (F4–F9) were characterized by a much higher content of aromatic compounds, polycyclic compounds, and condensation products, with a simultaneous decrease in the content of low-molecular non-oxygenated compounds (e.g., CO_2_, C1–C4 hydrocarbons). The presence of the latter was dominant only in the reference sample and the F1–F2 samples, indicating an insufficient degree of carbonization.

From the point of view of the use of biochar as an additive to cementitious mortar formulas used in thermal insulation systems, the VOC emission profile is of key importance. Volatile by-products, especially aldehydes and saturated hydrocarbons, can adversely affect the health of users and impair the properties of mortars by interacting with the cement matrix (e.g., slowing down hydration). Meanwhile, the more condensed, stable aromatic structures present in the fired biochar have lower chemical activity and lower emissions of volatile components, making it a more beneficial ingredient [38].

Therefore, biochar fired at ≥700 °C (i.e., F6–F9) seems to be the most suitable as a functional additive for cementitious mortars. Not only does it offer minimal VOC emissions, but it also potentially increases the durability, porosity, and thermal insulation properties of the composite material, which is desirable in external thermal insulation composite systems (ETICS).

Although this study primarily focused on the total VOC emissions from cement mortars containing biochar, it is important to recognize that molecular-level interactions may influence these emissions. Recent findings, such as those by Ali et al. [32], suggest that specific surface functionalities in engineered biochars (e.g., isolated Ca–Nx or Fe sites) can play a key role in adsorption or catalytic transformation of volatile compounds. These mechanisms—especially involving electron transfer and surface redox reactions—may be relevant for the future optimization of VOC emissions performance in cementitious composites. Although total VOC emissions were quantified using TIC profiles, specific identification of the emitted compounds was not conducted. Future studies will incorporate mass spectral interpretation and compound identification to better evaluate the toxicological relevance of individual VOC species.

### 3.11. Carbon Footprint and LCA Analysis

#### 3.11.1. Carbon Footprint for BC

To assess the environmental benefits of incorporating biochar into cementitious mixtures, comparative carbon footprint (CF) and LCA (A1–A3) analyses were conducted. Typical uncertainty range: ±5–15% of the total CO_2_-eq value.

The baseline reference sample, composed of traditional raw materials (cement, quartz sand, cellulose ether, and re-dispersible polymer powder), exhibited a carbon footprint of 29.96 kg CO_2_-eq per 100 kg of product, which corresponds to approximately 166.06 Mg CO_2_-eq annually based on a production volume of 554,275 kg/year (Table 4).

The following emission factors were used for the carbon footprint assessment (kg CO_2_-eq per kg of material):
Cement (CEM II 42.5R): 0.716 kg CO_2_-eq/kg [41].Sand (0–2 mm quartz): 0.005 kg CO_2_-eq/kg [42].Water (tap): 0.0003 kg CO_2_-eq/kg [43].Biochar: −1.00 kg CO_2_-eq/kg, based on net-negative values reported in the literature due to carbon sequestration potential [43].

Incorporation of biochar derived from biomass pyrolysis significantly reduced the carbon footprint of the resulting mixtures. Samples F3, F6, and F9, which contained 10% biochar (450 °C, 550 °C, and 700 °C, respectively), demonstrated a reduction of up to 34.78% in CF per 100 kg, with annual emissions lowered to 108.31 Mg CO_2_-eq/year.

This corresponds to a potential annual reduction of approximately 57.75 Mg CO_2_-eq when replacing a portion of the quartz filler with biochar. The assumed emission factor for biochar was −1.000 kg CO_2_-eq/kg, reflecting its net carbon sequestration potential. The reductions were consistent regardless of the pyrolysis temperature used for biochar preparation (Figure 6).

The environmental credit assigned to biochar in this study (−1.0 kg CO_2_-eq/kg) reflects both its carbon sequestration potential and its role in displacing emissions from conventional cement production. This value is supported by the existing literature, where biochar is widely recognized as a stable form of biogenic carbon with long-term storage potential in soils or cementitious matrices. Woolf et al. [44] demonstrated that sustainably produced biochar could mitigate global climate change by offsetting up to 12% of anthropogenic greenhouse gas emissions. The mitigation potential depends on multiple factors, including feedstock origin, pyrolysis conditions, and the durability of the application matrix.

Furthermore, as reviewed by Ravindiran, G. et al. [45], modification and engineering of biochar can enhance its physicochemical stability, sorption properties, and environmental functionality—further increasing its net environmental benefit when used in construction materials. In cementitious systems, where biochar becomes immobilized in the matrix, its long-term stability and low leaching profile provide a favorable environmental profile and support the allocation of a carbon credit in cradle-to-gate LCA models.

These findings confirm the high decarbonization potential of biochar as a sustainable additive in cementitious products, contributing to the transition toward climate-neutral building materials [46].

The results showed a clear reduction in the carbon footprint with the incorporation of biochar, particularly at higher pyrolysis temperatures and dosages. The reference sample exhibited the highest emission value of 166.1 Mg CO_2_/year, while the mix with 10% biochar showed the lowest footprint, at 108.3 Mg CO_2_/year. This represents a reduction of approximately 35% compared to the conventional mixture.

Such significant decreases in carbon emissions are attributed to both the substitution of energy-intensive cement and the carbon sequestration potential of biochar. These findings support the use of biochar as a promising additive in low-carbon cementitious materials.

#### 3.11.2. Life-Cycle Assessment (LCA)—Stages A1 to A3

A detailed life-cycle assessment (LCA) [25] was conducted for the reference cementitious sample and experimental formulations (F1–F9) modified with 1–10% biochar produced at varying pyrolysis temperatures (450 °C, 550 °C, and 700 °C). The analysis focused on cradle-to-gate stages: A1—raw material supply, A2—transport, and A3—production (Table 5).

The LCA results, expressed in Mg CO_2_-eq/year, reveal a clear trend: samples with increasing biochar content exhibit progressively lower environmental impacts across all stages. For the reference sample, total annual emissions reached 166.06 Mg CO_2_-eq/year, with A1 (raw materials) being the dominant contributor (99.64 Mg/year) (Figure 7).

In contrast, samples with 10% biochar reduced the A1 contribution to 64.98 Mg/year and total emissions to 108.31 Mg/year, representing a 34.8% reduction compared to the reference.

The stacked LCA breakdown (Figure 7) shows that reductions are primarily attributed to the replacement of mineral aggregates with biochar, which carries a negative carbon footprint due to its carbon sequestration capacity. Transport and production impacts (A2, A3) also decreased proportionally, although to a lesser extent.

The life-cycle assessment was conducted following current best practices in the construction industry, ensuring methodological alignment with recently published guidelines and recommendations [47].

These findings confirm the environmental viability of biochar incorporation, particularly at higher dosages (≥5%), as a strategy to significantly decarbonize cementitious construction materials.

## 4. Directions of Further Research and Development Prospects

Previous research results confirm that biochar (BC) can be a valuable component of cement composites, affecting both their performance and environmental footprint. However, to fully exploit its potential, it is necessary to continue and deepen research in several key areas:
Long-term durability of mortars with the addition of biochar. Aging tests should be carried out (frost resistance, resistance to humidity–temperature cycles, and chemical resistance) to assess the behavior of BC mortars in operating conditions for many years. It will be particularly important to determine the durability of thermal insulation and mechanical properties [47].Future studies will focus on comprehensive microstructural characterization of the cement–biochar interface using techniques such as SEM, MIP, and nanoindentation to better understand mechanical behavior at the micro-scale. Additionally, performance evaluation under dynamic loading conditions and long-term durability tests (e.g., freeze-thaw, carbonation, sulfate attack) will be conducted [48,49].Explore the potential of engineered biochars—that is, biochars modified or functionalized to tailor their physicochemical properties for specific applications in cementitious systems. Such modifications may include surface activation, mineral impregnation, or particle size optimization to improve compatibility with cement hydration processes, enhance pozzolanic activity, or control leaching behavior [50]. These tailored biochars could be designed to target particular performance criteria such as improved mechanical strength, lower permeability, or enhanced chemical durability in aggressive environments. Functionalization could also address current limitations, such as heavy metal mobility or water demand, making biochar more versatile and reliable across different types of cement matrices.

In particular, research could focus on surface oxidation or acid treatment to enhance bonding with cement hydrates, nano- or micro-scales to control dispersion and reactive surface area, co-impregnation with pozzolanic materials (e.g., silica, fly ash) to improve reactivity, and modifications aimed at reducing leachability of trace elements (e.g., Mo, As).

This approach opens pathways toward application-specific biochars, particularly relevant for structural concretes, repair mortars, and low-permeability grouts, aligning material performance with environmental and regulatory requirements.

Use of Alternative Types of Biomass. It is worth expanding the range of raw materials for BC production to include other lignocellulosic waste (straw, husks, reeds). This will allow for assessing the impact of biomass composition on final properties and identifying locally available, low-cost sources of BC [45].Indoor Air Quality Impact Assessment. The impact of BC content and type on indoor air quality should be investigated, especially in the context of VOC emissions under real-world conditions (e.g., emissions in rooms with high temperature and humidity) [39,46].Economics and Life-Cycle Cost Analysis (LCCA). It is recommended to extend the LCA to include a life-cycle cost assessment to estimate the actual viability of BC in building materials over a 20–50-year perspective, including production, transport, installation, and disposal costs [48]. The life-cycle assessment was conducted using a cradle-to-gate approach, with system boundaries including raw material acquisition, biochar production, and mortar formulation [49]. Impact categories were selected based on the EN 15804+A2 [50] standard and included global warming potential, abiotic resource depletion, and energy demand. Normalization was performed using the latest EU reference dataset (EF 3.0). A full uncertainty analysis was not included at this stage but will be incorporated in future work.Narrativization and Industrial Implementations. Biochar has significant potential as a component of future-proof building materials. However, its continued use requires an interdisciplinary approach, combining materials engineering, environmental chemistry, and life-cycle analysis. Expanding research in the proposed directions will enable the full implementation of this technology in construction practice, supporting the implementation of the goals of sustainable development and climate neutrality [51].

Future research will employ factorial experimental designs and statistical analysis (e.g., ANOVA) to independently assess the effects of pyrolysis temperature and biochar dosage on cement mortar performance.

Environmental follow-up will involve applying predictive transport models to simulate pollutant migration and refine risk assessment. To bridge laboratory findings with practice, pilot-scale trials of biochar-enhanced mortars will be implemented in prefabrication settings, with key performance indicators including workability, curing efficiency, bond strength, and carbon reduction per functional unit.

## 5. Conclusions

This research clearly confirmed that biochar can play a key role in the development of modern and ecological building materials.

The main objective was to assess the possibility of using biochar obtained from wood waste—fired at temperatures of 450 °C, 550 °C, and 700 °C—as a functional additive to thermo-modernization mortars. The aim was to investigate the effect of the type and quantity of biochar on the physicochemical properties of cement mixtures, their environmental impact (VOC emissions and metal leaching), and carbon footprint calculated by the LCA (life-cycle assessment) method.

The results showed that the pyrolysis temperature significantly affects the chemical composition and structure of biochar, resulting in differences in its performance as a building component. Biochar fired at 700 °C, thanks to its high carbon content and alkaline pH, improves thermal insulation and structural stability of mortars. Biochar 450 °C, on the other hand, offers improved sorption properties and chemical reactivity.

The use of BC in the amount of 1–5% allows for the improvement of mechanical parameters and adhesion, meeting the requirements of European standards (EAD, EN 998-1). However, at higher concentrations (>5%), deterioration of adhesion to substrates is observed. Studies on VOC emissions and the leaching of pollutants showed compliance with standards in most cases, although some samples (F0 and F7) exceeded the limits for selected elements, indicating the need for further monitoring.

From an environmental perspective, the most significant result was a reduction in the carbon footprint of mortars by up to 83% compared to the reference mixture, which confirms BC’s high decarbonization efficiency. This reduction was due to both the reduction in cement and sand consumption and the carbon sequestration potential of biochar.

Biochar can therefore have a multifunctional role: strengthening, insulating, and environmental. It can become a key ingredient of future-oriented, low-emission building materials, in line with the goals of the circular economy and the climate neutrality strategy. Future research should focus on long-term sustainability, impact on the microstructure of mortars, internal emissions, and the economic and normative evaluation of the technology.

## Figures and Tables

**Figure 1 materials-18-03941-f001:**
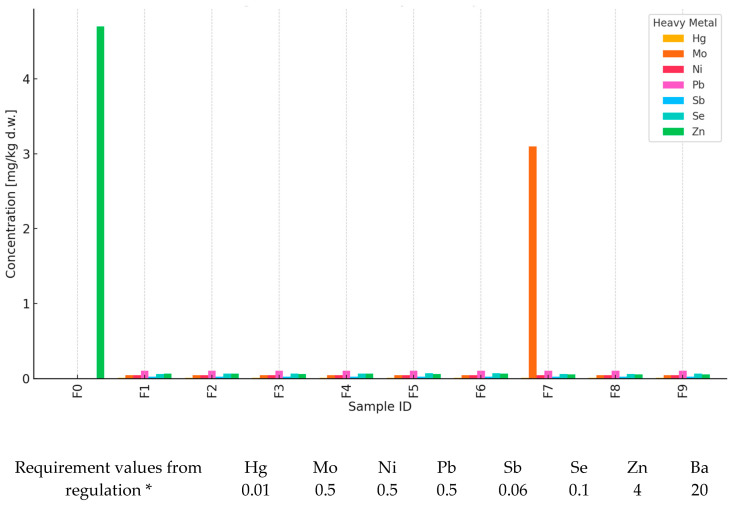
Analysis of heavy metal concentrations in waste samples. * Degree of leaching of elements (mg/kg) from mortars containing BC and comparison with the threshold values from Regulation of the Minister of Economy of 16 July 2015 [24] on the acceptance of inert waste at the landfill, Annex No. 2.

**Figure 2 materials-18-03941-f002:**
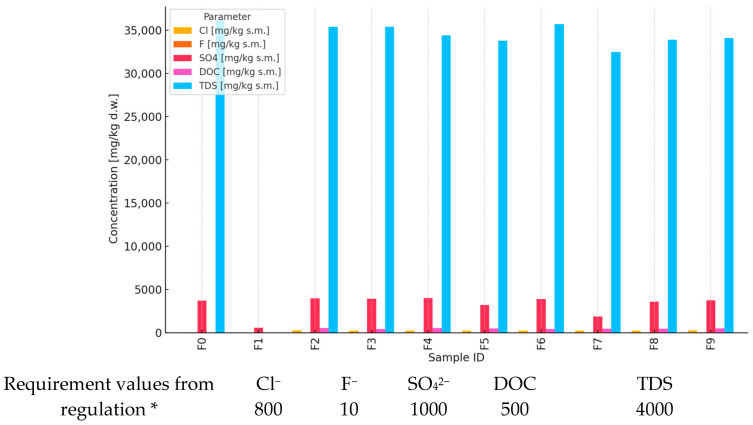
Analysis of inorganic salts and dissolved compounds in waste samples. * Degree of leaching of elements (mg/kg) from mortars containing BC (and comparison with the threshold values from the Regulation of the Minister of Economy of 16 July 2015 [12] on the acceptance of inert waste at the landfill, Annex No. 2).

**Figure 3 materials-18-03941-f003:**
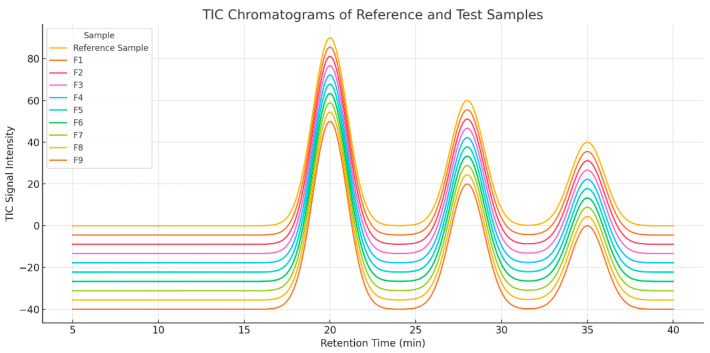
TIC chromatograms for reference sample F0 and samples with biochar content (F1–F9).

**Figure 4 materials-18-03941-f004:**
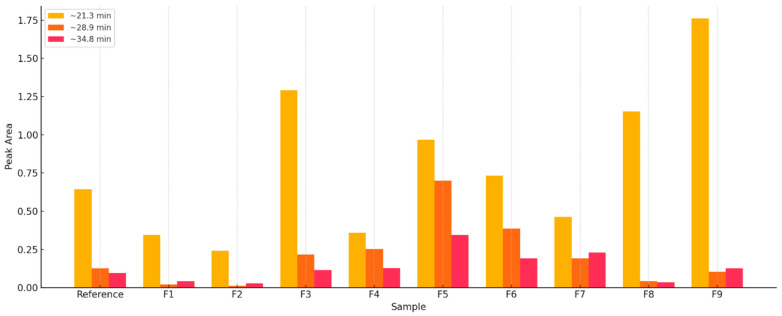
Relative peak area (%) for selected compounds by sample: F0, F1–F9.

**Figure 5 materials-18-03941-f005:**
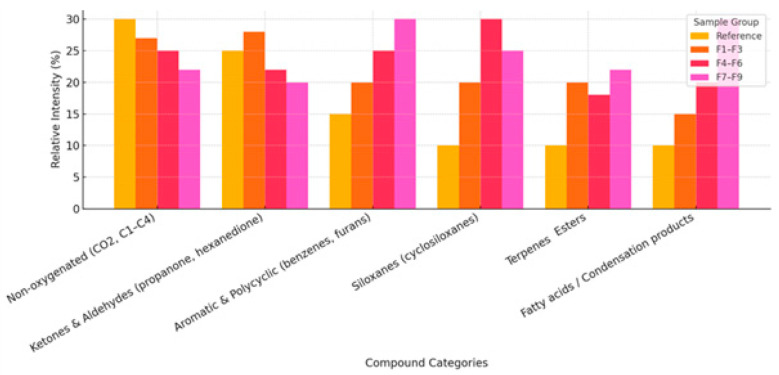
Relative abundance of chemical groups in biochar samples.

**Figure 6 materials-18-03941-f006:**
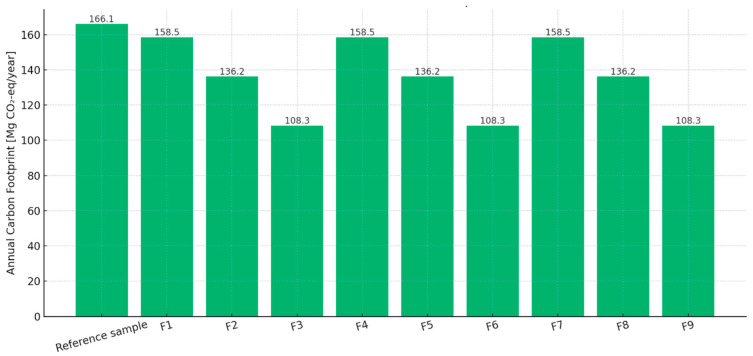
Annual carbon footprint per mix (Mg CO_2_/year).

**Figure 7 materials-18-03941-f007:**
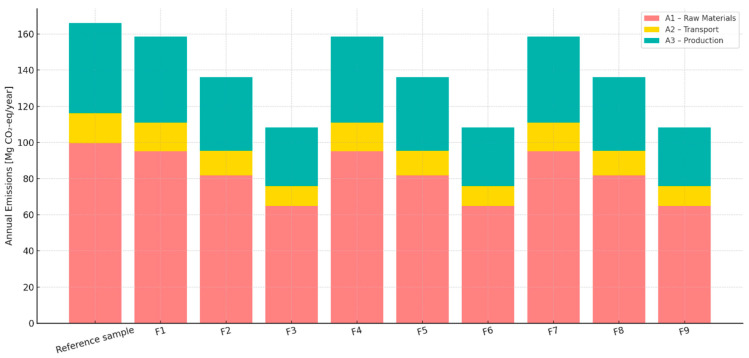
LCA (A1–A3) cradle-to-gate diagram for cementitious samples containing BC.

**Table 1 materials-18-03941-t001:** Formulation list, reference sample, and samples with the addition of biochar.

Ingredients [%]	Reference Formula	F1	F2	F3	F4	F5	F6	F7	F8	F9
**Cement mixture with additives**	68.7	67.7	67.7	67.7	63.7	63.7	63.7	58.7	58.7	58.7
**Biochar 450 °C**	-	1			5			10		
**Biochar 550 °C**	-		1			5			10	
**Biochar 700 °C**	-			1			5			10

**Table 2 materials-18-03941-t002:** Summary of test results for biochar.

Parameter	Unit	Biochar 450 °C	Biochar 550 °C	Biochar 700 °C
**Carbon content**	%	73.42	88.27	92.72
**Nitrogen content**	%	1.21	1.56	1.41
**Hydrogen content**	%	1.16	2.67	3.73
**Sulphur content**	%	0.44	0.85	0.97
**Oxygen content**	%	23.77	6.65	1.17
**pH**		6.6	7.0	9.0
**Ash content**	%	4.6	8.5	12.13

**Table 3 materials-18-03941-t003:** The results of the tests for the developed formulas.

Ingredients	Reference Formula	F1	F2	F3	F4	F5	F6	F7	F8	F9	Requirements
**Water/cement ratio w/c**	0.21	0.22	0.22	0.22	0.26	0.21	0.215	0.22	0.23	0.24	
**Consistency (Navikov)**	6.5	7	7	7	7	6.5	7	7	7	7	6.5–7.5
**EPS adhesion**	0.10	0.13	0.12	0.09	0.12	0.11	0.10	0.09	0.06	0.05	≥0.08 MPa
**Adhesion to concrete**	0.35	0.50	0.4	0.20	0.35	0.30	0.25	0.25	0.15	0.10	≥0.25 MPa
**Flexural strength**	6.4	5.5	5.5	5.2	4.6	6.2	6.2	6.15	6	5.7	Declared value
**Compressive strength**	25.5	24.6	23	21.8	20.5	22.1	23.8	25.5	25.3	24.8	(≥20 N/mm^2^) according to EN 998-2:2026

**Table 4 materials-18-03941-t004:** Annual carbon footprint for formulations developed.

Sample	Kg per 100 kg	Mg per Year
F0	29.96	166.06
F1	28.59	158.47
F2	19.54	158.47
F3	24.57	108.31
F4	28.59	136.19
F5	19.54	108.31
F6	24.57	108.31
F7	28.59	158.47
F8	19.54	108.31
F9	24.57	136.19

**Table 5 materials-18-03941-t005:** LCA A1–A3 analysis per sample.

Sample	A1	A2	A3
F0	17.98	3.0	8.99
F1	17.15	2.86	8.58
F2	14.74	2.46	7.37
F3	11.72	1.95	5.86
F4	17.15	2.86	8.58
F6	14.74	2.46	7.37
F5	11.72	1.95	5.86
F7	17.15	2.86	8.58
F9	14.74	2.46	7.37
F8	11.72	1.95	5.86

## Data Availability

The original contributions presented in this study are included in the article. Further inquiries can be directed to the corresponding author.

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
