# Peer review of "Biochar as a Sustainable Component of Low-Emission Building Materials"

_materials, 2025, doi:10.3390/ma18173941_

Round 1
Reviewer 1 Report
Comments and Suggestions for Authors
Originality
The Introduction section provides a general overview of biochar's advantages in construction but lacks a critical review of recent research advancements. This gap undermines the clarity of this study's innovative contribution. However, the multi-temperature gradient analysis (450°C/550°C/700°C) combined with comprehensive Life Cycle Assessment (LCA) demonstrates notable novelty.
Content and Analysis
Insufficient data interpretation:
Table 2 requires detailed analysis correlating biochar properties (e.g., ash content, pH) with performance metrics.
Environmental risk concerns:
F7 sample shows severe molybdenum leaching (3.1 mg/kg, 6× higher than the 0.5 mg/kg threshold). Causative analysis (e.g., feedstock contamination or high-temperature ash effects) must be provided.
F0 sample exceeds zinc limits (4.7 mg/kg vs. 4.0 mg/kg), indicating potential flaws in the reference formulation. Environmental implications should be reassessed.
LCA limitations:
The "cradle-to-gate" assessment omits key environmental impact categories (e.g., toxicity, eutrophication).
The biochar carbon footprint factor (-1.0 kg CO₂-eq/kg) lacks justification. Provide methodological references (e.g., IPCC guidelines) or sensitivity analysis.
Formatting and Data Issues
Figure/Table irregularities:
Figure 2 improperly combines graphical and tabular elements, reducing clarity.
Table 4 (annual carbon footprint) contains inconsistent column alignment and undefined abbreviations (e.g., "6H Sample").
Critical errors: Line 409 , Line 504 (undefined "F10"), and abstract punctuation require correction.
Referencing:
Ref. 17 is an invalid hyperlink. Replace with a functional source or archival identifier.
Overall formatting deviates from journal guidelines (e.g., inconsistent italics, author initials).
Data robustness:
All quantitative results (e.g., mechanical strength, leaching values) lack error margins/replication details. Report standard deviations and experimental replicates.
Author Response
Our responses are included in the attached file.
Thanks,
Andrzej Białowiec

Reviewer 2 Report
Comments and Suggestions for Authors
The manuscript entitled ""Biochar as a Sustainable Component of Low-Emission Building Materials"" presents an experimental investigation into the influence of pyrolysis temperature on the physicochemical, mechanical, and environmental performance of biochar-modified cementitious mortars. While the study addresses a timely topic and provides multi-dimensional testing results, it suffers from multiple critical issues in scientific rigor, experimental design control, mechanistic interpretation, and novelty. Based on these technical deficiencies, a "major revision" is required before the manuscript can be reconsidered for publication.
- Detailed physicochemical information of raw materials must be provided .
- The study omits characterization techniques such as SEM, MIP, or nanoindentation, which are vital to correlating biochar porosity and mortar strength.
- Although some relevant papers are cited, the review does not cover advanced uses of biochar in cementitious systems under dynamic loading or in multi-functional composites. For instance, “In-situ investigation on dynamic response of highway transition section with foamed concrete”may serve as a reference on dynamic response considerations.
- The manuscript relies exclusively on experimental data without incorporating predictive or mechanistic models. For a study dealing with VOC migration and leaching behaviors, the use of transport models (e.g., advection-dispersion equations) would enhance depth. The authors are recommended to refer to“Generalized solutions for advection-dispersion transport equations subject to time- and space-dependent internal and boundary sources” to improve analytical rigor.
- The discussion on VOC emissions lacks molecular-level insight into the sorptive interactions between cement phases and volatile species. The authors should incorporate analysis or discussion inspired by " The isolated Ca-Nx sites in biochar boosting Fe catalyzed Fenton-like oxidation of Tris(2-chloroethyl) phosphate: Properties, mechanisms, and applications", where surface reactivity and transition-metal sites are mechanistically analyzed.
- Only a basic cone-drop test is employed for consistency. Advanced rheometry (e.g., yield stress, thixotropy) would be more suitable to analyze the workability, especially given the porous nature of biochar.
- Tables 2–4 present overlapping information and are not self-explanatory. Some formulations (e.g., F4, F6) are inconsistently discussed in relation to their mechanical strength trends.
- The effects of pyrolysis temperature and dosage are entangled, making it difficult to isolate their independent contributions.
- While the LCA is mentioned, it lacks system boundary details, impact categories, normalization, and uncertainty analysis. The authors should expand on LCA in line with ISO and may consider referring to https://doi.org/10.1016/j.jclepro.2024.143681.Additionally, The adhesion tests to EPS and concrete are presented as bulk pull-off values without analyzing interfacial microstructure.
- Samples F0 and F7 showed significant leaching (e.g., Mo and Zn exceedances). This critical issue is downplayed and should be further elaborated through risk assessment frameworks.
- "ph" must be corrected to "pH" .
- . The conclusion section should not contain reference citations.
Author Response

(The authors gave the same response as above.)

Reviewer 3 Report
Comments and Suggestions for Authors
See attached document.

Author Response

(The authors gave the same response as above.)

Reviewer 4 Report
Comments and Suggestions for Authors
The manuscript provides a systematic evaluation of the feasibility of incorporating wood-derived biochar produced at varying pyrolysis temperatures (450 °C, 550 °C, 700 °C) into cementitious mortars for use in sustainable construction. The study comprehensively assesses physicochemical properties, mechanical performance, volatile organic compound (VOC) emissions, heavy metal leachability, and life cycle carbon footprint. The topic is timely and relevant, with clear implications for low-carbon materials and green building applications. The experimental framework is generally well designed, and the data is presented in a structured manner. However, several aspects related to experimental reproducibility, data interpretation, environmental compliance, and methodological transparency require improvement to enhance the manuscript’s scientific rigor and practical applicability.
*Section 2.2.2, Section 3 overall. The manuscript does not specify whether experiments were performed in replicates, nor does it report standard deviation values or apply statistical analyses to support data interpretation.
*Section 3.8.1. While biochar at 700 °C is promoted as the most effective additive, F7 shows molybdenum leachability exceeding permissible thresholds, which raises concerns about its environmental safety.
*Section 3.9. The VOC profile characterization lacks chemical identification of the compounds corresponding to major peaks in the TIC chromatograms, limiting the interpretability of their environmental or health impact.
*Section 3.10.1. The carbon footprint assessment omits detailed disclosure of the emission factors (e.g., kg CO₂-eq/kg for cement, sand, and biochar), which are critical for verifying the LCA outcomes.
*Section 3.7. The reduction in compressive strength for sample F4 (biochar at 550 °C, 1%) lacks discussion of potential causes such as material interaction or structural disruption, which is necessary for full evaluation.
*Section 4. The outlined directions for further research remain high-level and general, without describing methods, performance indicators, or implementation strategies that would guide practical follow-up.
Author Response

(The authors gave the same response as above.)

Round 2
Reviewer 2 Report
Comments and Suggestions for Authors
The authors have adequately addressed all technical issues through major revisions. However, the following corrections remain necessary:
Uncited reference claim: The response letter asserts alignment with "the latest LCA best practices in the construction industry" by referencing https://doi.org/10.1016/j.jclepro.2024.143681, yet this source is absent from the reference list. Incorrect source attribution: Reference [39] is inaccurately cited; the correct citation should attribute to Yang et al. The isolated Ca-Nx sites in biochar boosting Fe catalyzed Fenton-like oxidation of Tris(2-chloroethyl) phosphate: Properties, mechanisms, and applications, Applied Catalysis B: Environment and Energy, 2025, 366, 125056. "https://doi.org/10.1016/j.apcatb.2025.125056".
Author Response
Responses are attached in the file.

Reviewer 3 Report
Comments and Suggestions for Authors
The article has been substantially improved by the reviewers' suggestions. And with the changes introduced, the originality of the study has increased.
Author Response
Comment 1: The article has been substantially improved by the reviewers' suggestions. And with the changes introduced, the originality of the study has increased.
Response: Thank you for this comment. We appreciate it.
Reviewer 4 Report
Comments and Suggestions for Authors
The authors have satisfactorily addressed the main concerns raised in the first-round review, significantly improving the clarity of methods, data presentation, and overall structure. However, two minor issues related to references remain: (1) the reference list is not fully ordered according to the sequence of first appearance in the text renumbering is recommended; (2) the publication year format of Reference [40] does not conform to Materials journal standards and should be revised. These are all minor editorial issues and do not affect the scientific merit or publishability of the manuscript. Therefore, I recommend acceptance after minor adjustments during the final editing stage.
Author Response
Responses are attached in the file.
